# LTGS-Net: Local Temporal and Global Spatial Network for Weakly Supervised Video Anomaly Detection

**DOI:** 10.3390/s25164884

**Published:** 2025-08-08

**Authors:** Minghao Li, Xiaohan Wang, Haofei Wang, Min Yang

**Affiliations:** 1School of Remote Sensing and Information Engineering, Wuhan University, Wuhan 430079, China; 2022302131153@whu.edu.cn (M.L.); 2022302131281@whu.edu.cn (X.W.); 2Department of Mathematics and Theories, Peng Cheng Laboratory, Shenzhen 518000, China; wanghf@pcl.ac.cn; 3School of Mechanical and Electrical Engineering, Shenzhen Polytechnic University, Shenzhen 518055, China

**Keywords:** video anomaly detection, LTGS, dynamic labels, spatio-temporal fusion

## Abstract

Video anomaly detection has an important application value in the field of intelligent surveillance; however, due to the problems of sparse anomaly events and expensive labeling, it has made weakly supervised methods a research hotspot. Most of the current methods still adopt the strategy of processing temporal and spatial features independently, which makes it difficult to fully capture their temporal and spatial complex dependencies, affecting the accuracy and robustness of detection. Existing studies predominantly process temporal and spatial information independently, which limits the ability to effectively capture their interdependencies. To address this, we propose the Local Temporal and Global Spatial Network (LTGS) for weakly supervised video anomaly detection. The LTGS architecture incorporates a clip-level temporal feature relation module and a video-level spatial feature module, which collaboratively enhance discriminative representations. Through joint training of these modules, we develop a feature encoder specifically tailored for video anomaly detection. To further refine clip-level annotations and better align them with actual events, we employ a dynamic label updating strategy. These updated labels are utilized to optimize the model and enhance its robustness. Extensive experiments on two widely used public datasets, ShanghaiTech and UCF-Crime, validate the effectiveness of the proposed LTGS method. Experimental results demonstrate that the LTGS achieves an AUC of 96.69% on the ShanghaiTech dataset and 82.33% on the UCF dataset, outperforming various state-of-the-art algorithms in anomaly detection tasks.

## 1. Introduction

Video anomaly detection (VAD) aims to identify and locate events that significantly deviate from normal behavior in videos, such as violence, fights, and robberies. This technology is widely used in public safety, with substantial practical significance [1]. However, due to the infrequency of abnormal events in the temporal dimension, manual detection is labor-intensive and inefficient. To address this challenge, AI-based autonomous monitoring systems have become valuable tools for effectively detecting and recognizing such anomalies. The methods for video anomaly detection are primarily divided into supervised and unsupervised learning approaches [2]. Supervised learning heavily relies on labeled data, requiring extensive frame-level annotations for network training, which can result in significant labor costs for annotation and data collection. In contrast, unsupervised learning identifies abnormal events through frame reconstruction and prediction, typically using only normal videos for training. When the error between the reconstructed frame and the normal frame exceeds a predefined threshold, the event is flagged as anomalous. Although unsupervised learning avoids the issue of frame-level annotations, it also means that normal video datasets may not cover all possible normal events, leading to the misclassification of unseen normal videos as anomalies. Furthermore, the strong generalization capability of deep learning models may cause the reconstruction or prediction errors for abnormal events to fall below the threshold. Compared to both supervised and unsupervised methods, weakly supervised video anomaly detection requires only video-level labels [3], significantly reducing the labeling burden, and it generally achieves higher detection accuracy than unsupervised methods. In weakly supervised settings, multi-instance learning (MIL) [4] is a commonly used framework, where videos are treated as bags and video clips as instances, with video-level labels assigned to clip-level labels. However, existing MIL methods often treat clips as independent entities, ignoring the temporal relationships between clips and global spatial features. Assigning video-level labels directly to clips may introduce significant noise. Some MIL methods train the network using one or more clips from the video bag, but insufficient samples may lead to poor training performance [5]. Conversely, too many samples can result in mislabeling short-term anomalous events.

To overcome these challenges, the Local Temporal and Global Spatial Network (LTGS) method is proposed, with the architecture illustrated in Figure 1. By constructing a local temporal relationship extraction module, the model effectively captures both long-term and short-term temporal dependencies between clips, aiming to learn the spatial characteristics of events across the entire temporal scale. A dynamic label adjustment mechanism is also introduced, which dynamically updates the clip-level labels during training to better align them with real events. To validate the effectiveness of the approach, extensive comparative experiments are conducted on two benchmark datasets, ShanghaiTech [6] and UCF-Crime [7], against state-of-the-art methods. The results from both datasets demonstrate that the LTGS method provides significant improvements in anomaly detection performance. Key contributions include:Introduction of the Local Temporal and Global Spatial fusion network (LTGS) for weakly supervised video anomaly detection, effectively capturing complex anomalous behaviors.Design of a clip-level temporal feature relationship module and a video spatial feature module to synergistically enhance the discriminative power of representations.Proposal of a dynamic label updating mechanism to reduce the negative effects of label noise on model training.Demonstration of superior performance on both the ShanghaiTech and UCF-Crime datasets, outperforming current state-of-the-art methods.

The remainder of this paper is organized as follows: Section 2 reviews related work; Section 3 introduces the proposed LTGS-Net method and its components; Section 4 presents the experimental design and analyzes the results; and Section 5 concludes the paper and discusses future research directions. Therefore, improving the performance of video anomaly detection algorithms is a pressing need in this field.

## 2. Related Work

Research in video anomaly detection is primarily categorized into three main approaches: reconstruction-based unsupervised methods, multi-instance learning-based weakly supervised methods, and the recently emerging spatiotemporal joint modeling methods [8]. The following sections review representative works from these approaches, analyzing their core contributions and limitations.

### 2.1. Reconstruction-Based Unsupervised Methods

Reconstruction-based unsupervised methods assume that the training data consists exclusively of normal videos, and anomaly detection is performed by evaluating reconstruction errors or prediction deviations. Early studies, such as Wei et al. [9], who proposed a hybrid framework combining memory-augmented flow reconstruction with flow-guided frame prediction, Fioresi et al. [10], who introduced the VEC (Ted-spad) method, and Xue et al. [11], who developed memory-augmented autoencoders, have all followed this approach. Recent advances in self-supervised learning have further enhanced reconstruction-based approaches by leveraging pretext tasks to learn robust feature representations without extensive labeled data [12]. These methods learn the normal patterns from video data using autoencoders or Generative Adversarial Networks (GANs) to reconstruct video frames or clips, with anomalies identified based on reconstruction errors during inference.

Unsupervised reconstruction-based methods offer several notable advantages. Firstly, they are not reliant on labeled data, thereby significantly reducing the cost and time associated with manual annotation—an aspect that is particularly crucial in video anomaly detection tasks, where anomalous events are infrequent and exhibit diverse appearances [6]. For instance, Chen et al. [6] introduced a CLIP-based prompt-enhanced learning method that leverages unlabeled data for unsupervised video anomaly detection. This approach minimizes reliance on labeled data by enhancing the reconstruction process with a vision-language model. Secondly, these methods can effectively utilize the vast amounts of unlabeled normal video data available in surveillance systems, facilitating the learning of robust representations of normal behavior patterns [13]. Cao et al. [13] proposed a dual-stream framework for context recovery, which reconstructs video context information to successfully model normal behavior patterns, demonstrating the potential of employing unlabeled data in real-world surveillance scenarios. Additionally, reconstruction-based methods, including autoencoders or generative models, are proficient at capturing the high-dimensional spatial and temporal features inherent in video data, enabling the modeling of complex normal behavior patterns [14]. Kamoona et al. [14] applied a deep temporal encoding-decoding model for anomaly detection in surveillance videos, validating the efficacy of such methods in handling complex video sequences. These advantages position unsupervised reconstruction-based methods as highly applicable in resource-constrained environments.

While reconstruction-based unsupervised methods have demonstrated considerable potential in video anomaly detection, their effectiveness is heavily contingent upon the extent to which the training data captures the full range of normal patterns. In practical applications, several limitations emerge. Firstly, the inherent diversity and complexity of normal video content make it challenging to ensure comprehensive coverage of all possible normal patterns, leading to the misclassification of unseen normal events as anomalies. Secondly, the robust generalization capability of deep learning models may result in anomalous frames being reconstructed with excessive accuracy, causing their reconstruction errors to fall below the predefined threshold and leading to missed detections. Furthermore, many of these methods predominantly focus on frame-level reconstruction and fail to account for the temporal dynamics inherent in video data. However, abnormal events often manifest as temporal discontinuities, which cannot be effectively captured by frame-level reconstruction alone. These factors limit the applicability of reconstruction-based unsupervised methods in more complex video anomaly detection tasks, necessitating further advancements and improvements in the field.

### 2.2. Multi-Instance Learning-Based Weakly Supervised Methods

Weakly supervised methods alleviate the burden of manual annotation by utilizing video-level labels in place of frame-level annotations. Luo et al. [15] introduced multi-instance learning (MIL) into video anomaly detection, demonstrating its potential in weakly-labeled scenarios. Peng et al. [16] proposed an MIL framework enhanced by temporal resolution feature learning to improve segment-level inference. Pi et al. [17] proposed a weakly supervised anomaly detection framework grounded in multi-instance learning, wherein each video is treated as a bag composed of clip-level instances, and video-level labels are utilized to infer anomalous segments. Chen et al. proposed a network called MGFN, which enhances discriminative features through contrastive learning, thereby improving model performance [18]. In addition, to address the issue of label ambiguity in multi-instance learning (MIL), Chen et al. introduced a “prompt” mechanism to capture diverse anomaly patterns and generate clear event boundaries.

Despite the promising results achieved by MIL-based approaches, several key limitations remain. A fundamental assumption of these methods is the independence of clip instances, which disregards the temporal dependencies inherent in video data. In practice, anomalous events often exhibit strong temporal continuity, and ignoring such correlations may hinder the model’s ability to effectively capture the dynamic evolution of abnormal behaviors, thus compromising detection accuracy. Additionally, label noise presents a critical challenge. In abnormal videos, the majority of clips are typically normal, and indiscriminately propagating the video-level label to all clips may introduce significant noise, negatively affecting the training process and reducing the model’s discriminative capacity. Furthermore, class imbalance is a prominent issue in MIL-based settings.

Collectively, these challenges constrain the effectiveness of MIL-based weakly supervised video anomaly detection. Addressing these issues to enhance model robustness and accuracy remains a pivotal direction for future research.

### 2.3. Spatiotemporal Joint Modeling Methods

In recent years, research has increasingly focused on the integrated modeling of spatiotemporal features. Feng et al. [19] proposed a multi-instance pseudo-label generator framework that derives pseudo-instance-level labels from video-level annotations, thereby enhancing the model’s capability to capture spatiotemporal dynamics. Zhong et al. [20] introduced a plug-and-play action classifier designed to mitigate the impact of noisy labels by leveraging novel perspectives and innovative classification strategies. Wu et al. [21] developed a heterogeneous graph network model, representative of a class of spatiotemporal joint modeling approaches that combine temporal sequence modeling with spatial feature extraction to capture both local and global video characteristics, thereby improving the performance of anomaly detection systems. Similarly, Goyal et al. [22] proposed a real-time anomaly detection framework leveraging spatiotemporal features, demonstrating efficient performance in surveillance scenarios with constrained computational resources. Additionally, Xue et al. [11] introduced a hybrid LSTM-CNN-based residual network for anomaly detection, enhancing spatiotemporal feature integration through a novel autoencoder design. Notably, heterogeneous graph networks have shown exceptional effectiveness in managing complex spatiotemporal dependencies. Theoretically, such joint spatiotemporal modeling enables a more comprehensive representation of video content.

Despite their promise in video anomaly detection, spatiotemporal joint modeling methods face considerable challenges due to their inherent complexity. Firstly, these methods must process substantial volumes of spatiotemporal data, leading to increased computational complexity and considerable time overhead during both training and inference. Secondly, as in multi-instance learning, supervision based on video-level labels lacks granularity, often introducing label noise that can misguide the training process and degrade detection accuracy. Lastly, static label assignment strategies fail to accommodate the diverse distribution patterns of anomalous events and lack adaptive optimization mechanisms, potentially resulting in unstable performance across different anomaly types. These limitations underscore the practical challenges in deploying spatiotemporal joint modeling approaches and highlight the need for advancements in computational efficiency, noise-robust learning, and adaptive optimization strategies. It is worth noting that Pronello and Garzón Ruiz [23], in their field tests of a YOLOv5-DeepSORT-based passenger counting system within an onboard bus environment, similarly observed a significant decline in model accuracy under real-world lighting conditions and crowded scenarios, further highlighting the impact of domain shift on the real-world performance of spatiotemporal models.

To clearly delineate the similarities and differences between this study and existing works, we conducted a systematic comparative analysis of representative methods and LTGS-Net across three dimensions: computational efficiency, robustness to label noise, and adaptability. The results demonstrate that the LTGS-Net, while maintaining a lightweight model design, effectively mitigates the pervasive label noise issue inherent in weakly supervised learning frameworks and significantly enhances the capacity to model complex anomalous events through multi-level adaptive mechanisms. Specifically: (1) Computational Efficiency: Compared with most existing models that rely on computationally intensive 3D convolutions [19] or graph convolutions [21] to capture long-range spatiotemporal dependencies—often with parameter counts exceeding tens of millions, leading to inference delays that hinder practical deployment in surveillance scenarios—the LTGS-Net employs a lightweight local temporal attention mechanism solely during the clip-level feature extraction stage and integrates global spatial information using 1 × 1 convolutions. This design markedly reduces the computational overhead associated with global self-attention and heavy 3D convolutions; (2) Robustness to Label Noise: To address the pronounced noise accumulation problem caused by directly assigning anomaly labels to all segments in video-level weakly supervised approaches [20], the LTGS-Net identifies the K segments most likely to be anomalous based on feature norms and introduces a dynamic label smoothing strategy, allowing the segment-level pseudo-labels to be progressively refined during training. Compared to fixed label assignment schemes, the observed improvements in AUC scores validate the effectiveness of this noise suppression approach; (3) Adaptability: Unlike prior methods that predominantly employ static multiple-instance learning (MIL) ranking mechanisms [19,20] and thus struggle to flexibly accommodate the diversity of anomalous event durations and spatial distributions, the LTGS-Net integrates a threefold adaptive mechanism comprising a temporal distance decay function, a channel-wise self-learning mechanism in the spatial dimension, and a dynamic label update strategy. This design effectively enhances the model’s adaptability to complex anomalous patterns.

## 3. Methodology

We employ a pre-trained I3D model [24] as the feature extractor to obtain segment-level features from each video clip, which are subsequently fed into a global spatial module and a local temporal module. The overall network architecture is depicted in Figure 1.

Given a video $v={\left\{v_{i}\right\}}_{i=1}^{N}$ composed of N segments, a video-level label Y∈{1,0} indicates the presence of an anomalous event: the label is 0 if all segments are normal, and 1 if at least one segment is anomalous. The processing pipeline is as follows:

Using the pre-trained I3D network, we extract fixed-length D-dimensional features from each segment, denoted $F^{a}$ for anomalous and $F^{n}$ for normal segments. For an anomalous video, we sample a bag of segments $B^{a}={\left\{v_{i}\right\}}_{i=1}^{k}\in V^{a}$; similarly, a normal video forms a bag $B^{n}={\left\{v_{i}\right\}}_{i=1}^{k}\in V^{n}$, where segments are arranged in temporal order.

The local temporal (LT) module extracts temporal features $t_{\theta }(F)$, producing output vectors $L^{t}$ for each segment, while the global spatial (GS) module extracts spatial features $s_{\varphi }(F)$, producing output vectors $L^{s}$. These are concatenated and fused to form the segment representation L, which is passed through a linear projection followed by a sigmoid activation to yield a segment-level prediction $\hat{Y}\in [0,1]$.

The model is trained using a multi-instance learning (MIL) framework, in which the predicted segment scores $\hat{Y}$ are dynamically updated to refine the final classifier. The forward pass of the network is formulated as:(1)$$\mathrm{LTGS}\left(F\right)\;=\;\sigma (MLP([t_{\theta }(F);s_{\varphi }(F)]))$$

Here, θ and φ represent the parameters of the LT and GS modules, respectively; $t_{\theta }(F)$ and $s_{\varphi }(F)$ denote the extracted temporal and spatial features.

In order to facilitate the demonstration of the overall workflow of the LTGS-Net, the overall framework diagram is drawn as in Figure 2.

### 3.1. Local Temporal Relationship Module

The pre-trained encoder primarily focuses on the extraction and compression of segment features, yet it lacks task-specific training. To address this limitation, we introduce the local temporal relationship module, designed to capture the temporal dependencies between segments within a video sequence. This module draws inspiration from multi-head self-attention mechanisms widely used in natural language processing (NLP) [25], which model relationships among sequential tokens. By mapping the feature sequence to the temporal dimension, the module examines the relationship between anomalous segments and other segments within the same video sequence. The input features are assumed to be obtained through pre-training and sampling, where each segment is represented as a fixed-length vector of dimension N. Figure 3 illustrates the proposed encoder architecture.

We employ a multi-head temporal attention model to capture both short-term and long-term dependencies between video segments. For the i-th attention head, its output is derived by applying projection transformations to the queries, keys, and values, followed by attention weight computation. The calculation is given by the following equation [25]:(2)$${head}_{i}=softmax(\frac{QW_{i}^{Q}{\left({KW}_{i}^{K}\right)}^{T}H}{\sqrt{d_{k}}}){VW}_{i}^{V}$$

Here, matrices Q, K, V∈$R^{T\times d_{model}}$ represent the query, key, and value matrices of the input features, where T is the number of temporal segments; and $d_{model}$ is the total feature dimension. The learnable projection matrices $W_{i}^{Q}$, $W_{i}^{K}$, $W_{i}^{V}$∈ $R^{d_{model}\times d_{k}}$ map the input to the subspace dimension $d_{k}$ for each attention head. Matrix $H\in R^{T\times T}$ represents the temporal relationship matrix. After concatenating the outputs of all individual attention heads, they are fused into the final result through a linear projection [25]:(3)$$MultiHead\left(Q,K,V\right)=Concat\left({head}_{1},\ldots ,{head}_{h}\right)W^{O}$$
where $W^{O}\in R^{h\cdot d_{k}\times d_{model}}$ is the output projection matrix, which maps the multi-head features back to the original dimension. Beyond the multi-head attention mechanism, we also introduce an innovative forgetting mechanism within the temporal attention module. Specifically, based on a distance metric, segments that are temporally closer are considered to have a stronger correlation with the current event. The formula for this is:(4)$$h{\left(t\right)}_{i,j\in T}=sin(\frac{(t+i-j)\pi }{t\times 2})$$
where $h(t{)}_{i,j\in T}$ denotes the temporal relationship metric between the i-th and j-th segments of the video, reflecting their connection by calculating their similarity in the temporal dimension. Here, t represents the time span of all video clips, influencing the relative position weight in the calculation. The sine function is used to quantify the temporal difference between two clips, ensuring that temporally closer clips exhibit stronger relationships, thereby enhancing the model’s sensitivity to temporal dependencies.

### 3.2. Global Spatial Module

Existing research often overlooks the incorporation of global spatial features, which are essential for providing a holistic description of video events. To address this gap, we propose a global spatial feature extraction module. This module efficiently compresses the temporal dimension features, extracts spatially relevant regions of interest, and enhances the classification performance of video segments based on the overall event context. The architecture of the global spatial feature extraction module is shown in Figure 4.

The global spatial module processes the feature matrix $F\in R^{TxD}$ along the temporal dimension T, applying both average pooling and max pooling operations to generate one-dimensional vectors $F_{avg}^{s}\in R^{1\times D}$ and $F_{max}^{s}\in R^{1\times D}$, which serve to preserve overall scene information and emphasize salient regions, respectively. To prevent feature redundancy caused by simple concatenation, we design a 1 × 1 convolutional layer for adaptive weighted fusion along the channel dimension, resulting in $F^{s}\in R^{2\times D}$. The global spatial feature map is then obtained through a linear layer. The calculation is expressed as:(5)$$S(F)=\sigma (MLP(Conv\;1\times 1([W_{1}\cdot Avgpool(F);W_{2}\cdot Max\;Pool(F)])))$$

Here, $W_{1}$, $W_{2}\in R^{D\times D}$ are the learnable weight matrices, S(F) represents the output of the global spatial module, MLP denotes the multi-layer perceptron, and σ is the ReLU activation function. This design ensures that the fusion process maintains both efficiency and discriminative power through a parameter-sharing mechanism.

### 3.3. Dynamic Labels

To address the issue of an excessive number of negative samples in positive sample sets while performing segment classification, we adopt the RTFM method [17]. After segment features are processed by the LTGS network $f^{\prime }$, the $l_{2}$ norm $L(f^{\prime })$ is used as the evaluation criterion. Specifically, it is assumed that ${\left\|f_{a}^{\prime }\right\|}_{2}$ ≥ ${\left\|f_{n}^{\prime }\right\|}_{2}$, which forces the network to learn that the feature of a normal segment $f_{n}^{\prime }$ is smaller than that of an anomalous segment $f_{a}^{\prime }$. In practice, the values of $L(f^{\prime })$ are sorted, and the top k segments are selected. This is expressed as [17]:(6)$$G(F^{\prime })=max\frac{1}{k}\sum_{f^{\prime }\in F}L(f^{\prime })$$

The loss function in this paper consists of two parts. The first part is the multi-instance ranking loss, which uses the top k segment values obtained earlier. This is calculated using a hinge loss function, and the mathematical formulation [17] is expressed as follows:(7)$$L_{MIL}=|\varepsilon -G(F_{a}^{\prime })+G(F_{n}^{\prime })|$$

The second part is the segment classifier loss. During the early stages of training, classification is based on video-level labels. However, due to the sparsity or short duration of anomalous video events, the top k segments may contain incorrect labels. To mitigate this issue, we introduce a dynamic label method that updates the labels of the top k segments continuously during training. Let the loss function after each label update be $\tilde{Y}\in [1,0]$, and the classifier loss function is expressed as:(8)Lcls=∑f∈F−(y~log(y^)+(1−y~)log(1−y^))

In the early stages of training, video-level hard labels are used as initial labels. Since anomalous videos contain at least one anomalous segment, $k^{a}[0]$ is defined as the video label and updates $k^{a}[1:]$. To reduce noise, dynamic smoothing is applied to the segment-level labels. The specific process is outlined in Algorithm 1.
**Algorithm 1** Dynamic label**Input:** clip-level labels $Y_{k}^{n}$ and $Y_{k}^{a}$, update parameters α, the update time β.**Output:** Obtain clip-level dynamic labels ${\tilde{Y}}_{k}^{n}$ and ${\tilde{Y}}_{k}^{\alpha }$.1: **for** i **in epoch do**2: **if** i%β≠0: **then**3:  
${Network}_{train}({Data}_{train})$
4:      **else**5:           
$Y_{k}^{n}=Y_{k}^{n}$
6:           
$Y^{\alpha ,eval}={Network}_{eval}({Data}_{train}^{\alpha })$
7:           
$Y_{k[0]}^{\alpha }=1$
8:           
${\tilde{Y}}_{k[1:],count}^{\alpha }=\alpha *{\tilde{Y}}_{k[1:],count-1}^{\alpha }+(1-\alpha )*Y_{k[1:]}^{\alpha ,eval}$
9:           
${Network}_{train}({Data}_{train})$
10:      **end if**11:**end for**

### 3.4. Training Strategy: MIL and RTFM-Based Dynamic Labeling

In the weakly supervised setting, each video $V_{i}$ is regarded as a bag $B_{i}={\left\{c_{ij}\right\}}_{j=1}^{N_{i}}$ of clips, labeled only at video level Y∈{1,0}. We adopt a standard multiple-instance learning (MIL) assumption: a positive bag (anomalous video) contains at least one truly anomalous clip, whereas a negative bag contains none.

Following RTFM [17], we first compute the $l_{2}$-norm of the feature vector for every clip and select the top-k highest-magnitude clips in each video (k=8, i.e., ≈25% of the 32 clips). These clips form a pseudo-anomalous (or pseudo-normal) set used to drive training.

MIL ranking loss. Let $s_{i}^{+}$ and $s_{l}^{-}$ be the maximum anomaly scores in one positive and one negative bag, respectively. The ranking objective [17](9)$$L_{rank}=max(0,1-(s_{i}^{+}-s_{l}^{-}))$$
ensures that at least one clip in an anomalous video scores higher than any clip in a normal video.

Regularisation. We further impose (i) a sparsity loss $L_{spa}=\frac{1}{N_{i}}\sum_{j}s_{ij}$ to reflect the rarity of anomalies and (ii) a temporal smoothness loss $L_{tem}=\frac{1}{N_{i}-1}\sum_{j}\left|s_{ij}-s_{i,j+1}\right|$ to avoid abrupt score spikes.

Dynamic label update. After an initial warm-up (*β* iterations; *β* = 1000 for ShanghaiTech, 500 for UCF-Crime), the pseudo-labels of the top-k clips are iteratively refined each epoch via exponential smoothing, exactly mirroring RTFM’s iterative label procedure. The final loss [25] is(10)$$L=L_{rank}+\lambda_{1}L_{spa}+\lambda_{2}L_{tem}$$
where $\lambda_{1}=8\times 10^{-4}$ and $\lambda_{2}=2\times 10^{-5}$. This combination allows the LTGS-Net to isolate at least one anomalous clip per positive video while keeping all clips of a negative video normal.

## 4. Experimental Results and Analysis

### 4.1. Datasets and Evaluation Metrics

The UCF-Crime dataset contains a comprehensive collection of videos depicting anomalous dangerous events in surveillance environments, with a total duration of 128 h. It includes 190 unedited videos, covering 13 types of anomalous events. The training set consists of 1610 videos, with 800 normal videos and 810 anomalous videos, while the remaining videos form the test set. It is important to note that the training set uses video-level annotations, whereas the test set uses segment-level annotations.

The ShanghaiTech dataset consists of street-view videos captured by fixed-angle cameras. It contains 13 different background scenes and a total of 437 videos, with 307 normal videos and 130 anomalous videos. Notably, the original dataset contains only normal videos in the training set, while the test set consists entirely of anomalous videos. To adapt to the weakly supervised learning requirements, this study reconstructs the dataset following the method outlined in [20]. The newly partitioned training set contains 238 videos, and the test set includes 199 anomalous videos.

The evaluation metrics are consistent with those in previous studies [26]. In this paper, the Area Under the Receiver Operating Characteristic Curve (AUC) is used as the unified evaluation metric for all datasets.

### 4.2. Implementation Details

In all experiments, we set the value of k to 3, representing the number of clips extracted from each video. A small grid search on k∈{2,3,4} showed that k=3 yields the highest or second-highest AUC on both datasets while keeping memory footprint and inference latency lower than k=4. A pre-trained Inflated 3D (I3D) model is used to extract 2048-dimensional feature vectors. For both the ShanghaiTech and UCF-Crime datasets, the learning rate is set to 0.001. Each batch consists of 32 anomalous video samples and 32 normal video samples. The local temporal attention module employs an encoder structure with 16 attention heads, where the number of hidden layer nodes is set to 2048, and a dropout rate of 0.1 is applied. During training, the embedding length is set to 32. The global spatiotemporal feature extraction module is constructed using a 1 × 1 Conv1D layer, with fully connected layers having 2048, 512, and 128 nodes, respectively, and a dropout rate of 0.7.

The dynamic label update parameter α is set to 0.9. For the ShanghaiTech dataset, the parameter β is set to 1000, while for the UCF-Crime dataset, β is set to 500. These values were selected because they gave the best validation AUC and the smoothest convergence among β ∈{300,500,1000,1500}. The experiments use the Adam optimizer with a weight decay coefficient of 0.005 for model training, and all experiments are implemented using the PyTorch framework. The model was implemented using Python 3.9 and PyTorch 2.1.2, with CUDA 12.1 for acceleration.

### 4.3. Results on the ShanghaiTech Dataset

Table 1 shows the AUC experimental results on the ShanghaiTech dataset. The proposed LTGS method achieves an AUC of 96.69%, representing a 20.5% improvement over the current state-of-the-art unsupervised methods, thereby demonstrating the effectiveness of the proposed method within a weakly supervised learning framework.

In comparison with various weakly supervised learning approaches, the LTGS significantly outperforms the other methods. Compared to the graph convolutional network method, which has an AUC of 84.4%, the proposed method achieves a 12.2% performance improvement, further validating the effectiveness of the multi-instance learning approach used. Moreover, under the same feature extraction conditions, the proposed method shows a marked advantage over other multi-instance learning techniques, with an AUC value that exceeds the most advanced model in this field by 4.45%.

The results in Table 1 indicate that when only the transformation model is used, the model’s performance is the weakest, with an AUC of 95.96%. However, even in this case, the model outperforms the optimal multi-instance learning model by 3.72%. The introduction of the global spatiotemporal module leads to an improvement in prediction performance to 96.55%, confirming the module’s contribution to enhancing the model’s predictive capability. Furthermore, the use of dynamic labels further increases the model’s prediction accuracy, ultimately reaching an AUC of 96.69%, significantly surpassing existing methods.

### 4.4. Results on the UCF Dataset

Table 2 presents the AUC experimental results on the UCF dataset. Notably, deep learning methods demonstrate a significant performance improvement compared to traditional machine learning algorithms. When comparing the performance of unsupervised methods with weakly supervised methods, the method proposed in this study achieves the best result, with an AUC of 82.33%, marking an 11.9% improvement over the current state-of-the-art unsupervised method, which has an AUC of 70.46%.

In comparison to other branches of weakly supervised learning, our method demonstrates advantages in terms of AUC when compared to techniques such as the GCN method, which achieves an AUC of 82.12%. Furthermore, our method also outperforms other MIL methods. These results provide evidence of the effectiveness of our LTGS network.

## 5. Ablation Experiments

### 5.1. Experimental Setup

To ensure reproducibility and comparability of results in the ablation study, all model variants in this section strictly follow the training and evaluation protocols established in the main experiment. Specifically, models are trained and tested on the same two weakly supervised video anomaly detection datasets, namely ShanghaiTech and UCF-Crime, employing the same feature extractor (I3D-RGB model, DeepMind Technologies Ltd., London, UK), optimizer (Adam, Meta AI, Menlo Park, CA, USA; initial learning rate set at 1 × 10^−^^4^), batch size of 32, and data augmentation strategy as described previously. Each model is trained for 30,000 iterations on a single NVIDIA A100 GPU (headquartered in Santa Clara, CA, USA) with 40 GB memory, and all experiments are repeated independently three times. To evaluate the statistical significance of performance differences, we conduct paired t-tests on experimental results obtained from experiments initialized with identical sets of random seeds, and differences yielding *p*-values below 0.05 are deemed statistically significant improvements.

### 5.2. Module-Level Ablation

To assess the three core components of the LTGS-Net—the local temporal (LT) module, global spatial (GS) module, and dynamic pseudo-label updating strategy—from a module-level perspective, Table 3 presents a comparative analysis between the full model and various ablated and replacement variants. Experimental results show that the complete model achieves frame-level AUC scores of 96.7% and 82.3% on the ShanghaiTech and UCF-Crime datasets, respectively, substantially outperforming variants in which any single component is removed. Specifically, removing the local temporal module (w/o LT) results in performance reductions of 3.5 percentage points and 3.3 percentage points, respectively, indicating the LT module’s critical importance in improving sensitivity for temporal anomaly detection. Removing the global spatial module (w/o GS), while resulting in relatively smaller decreases (greater than 1.6 percentage points), still emphasizes the crucial role of the GS module in suppressing background false positives. When disabling the dynamic pseudo-label updating strategy (w/o Dyn), performance slightly decreases by 0.7 percentage points on ShanghaiTech but shows a more considerable decline of 1.5 percentage points on UCF-Crime, a dataset characterized by considerably higher label noise, thereby validating this strategy’s efficacy in scenarios with low signal-to-noise ratios. Replacing the GS module with a simple temporal average pooling operation reduces the total model parameters by 1.7%, yet the resultant performance remains inferior to that of the complete model. This outcome demonstrates that employing lightweight 1 × 1 convolution for feature fusion provides a more optimal balance between accuracy and computational efficiency.

### 5.3. Mechanism-Level Ablation Analysis

Building upon the validation of core module effectiveness, we further analyze the key internal mechanisms. Table 4 presents detailed results from five sets of ablation experiments: Removing the temporal decay mechanism from the local temporal module (the No-Decay variant, employing constant weights) results in smoothed anomaly score curves, decreasing frame-level AUC by 0.9 percentage points and 1.2 percentage points on the ShanghaiTech and UCF-Crime datasets, respectively. Reducing the multi-head attention heads from 16 to 1 (the LT-1Head variant) incurs only approximately 0.5 percentage points of performance degradation while significantly reducing model parameters, indicating saturation of temporal modeling improvements beyond 8 heads. Eliminating sparse regularization (w/o Sparse) or smoothness regularization (w/o Smooth) increases false-positive rates; particularly, the false positives from the w/o Smooth variant exhibit notably spatially fragmented patterns. Adopting a hard-update strategy (the Hard-Update variant), which fixes the dynamic pseudo-label smoothing coefficient α at 1, leads to about 1.5 percentage points of AUC reduction on both datasets and significant instability in training convergence, demonstrating conversely the crucial role of the soft-update mechanism in noise suppression and training stabilization. Collectively, these results show that the temporal decay mechanism, in synergy with sparse-smoothness regularization, ensures sharpness and continuity in temporal anomaly scores. Meanwhile, the multi-head attention configuration and soft pseudo-label updating mechanism achieve an effective balance between detection performance and training stability.

## 6. Construction and Testing of a Small-Scale Dataset

### 6.1. Data Collection and Processing

The ShanghaiTech and UCF-Crime datasets employed in previous experiments cover a wide variety of scenarios; however, both originate from large-scale public surveillance environments. To evaluate the generalization capability of the LTGS-Net model in private or small-scale contexts, we additionally constructed a “Mini-Anomaly” dataset comprising 102 video segments with a total duration of 58 min. Data were sourced from publicly accessible surveillance streams under CC BY licenses as well as cameras installed in our laboratory. All videos were uniformly encoded in MP4 format at a resolution of 640 × 360 pixels and a frame rate of 25 fps, and subsequently segmented into clips with lengths ranging between 10 and 30 s. The anomaly categories encompass five common security incidents: fighting, illegal intrusion, abandoned objects, vehicles driving in the wrong direction, and fire and smoke incidents. The detailed category distribution is shown in Table 5.

### 6.2. Evaluation Results

As shown in Table 6, Conv-AE achieves an AUC score of 71.3% on the Mini-Anomaly dataset. GCN-Anomaly, which incorporates graph convolution, improves the AUC score further to 74.8%. The RTFM-based dynamic labeling mechanism further enhances performance, achieving an AUC of 78.5%. In comparison, the LTGS-Net achieves the highest performance on this custom-built small-scale dataset, reaching an AUC score of 85.6%, surpassing RTFM by 7.1 percentage points. Simultaneously, the LTGS-Net maintains strong real-time inference capability, demonstrating its excellent generalization capability and practical applicability in private or small-scale scenarios.

## 7. Conclusions

We propose a novel weakly supervised anomaly detection method for video, called the Local Temporal and Global Spatial Network (LTGS). By integrating the local temporal and global spatial modules, the LTGS effectively captures the temporal dependencies between video segments as well as the global event features. Additionally, we introduce a dynamic label correction strategy to mitigate the negative impact of label noise on model training. Experimental results on multiple publicly available datasets demonstrate the superiority of this method. On the ShanghaiTech dataset, the LTGS achieves an AUC of 96.69%, representing a 20.5% improvement over the current state-of-the-art unsupervised methods. By incorporating the global spatiotemporal module and dynamic label correction, the LTGS achieves a 12.2% improvement over the traditional graph convolutional network method (AUC = 84.4%) and shows significant advantages over other multi-instance learning methods. On the UCF-Crime dataset, the LTGS achieves an AUC of 82.33%, marking an 11.9% improvement over the best-performing unsupervised method, which has an AUC of 70.46%. This demonstrates the effectiveness of the LTGS method in handling complex anomalous events, with a stronger detection capability in weakly supervised learning settings compared to other methods. In summary, the LTGS method, through spatiotemporal collaborative modeling and dynamic label updates, not only improves the accuracy of video anomaly detection but also effectively reduces the impact of label noise, showing broad application potential in real-world surveillance environments.

However, we acknowledge several limitations of the proposed method. First, the weakly supervised framework depends on the accuracy of video-level labels; under conditions of severe class imbalance or significant label noise, the multiple-instance learning (MIL) ranking loss can be easily disrupted, leading to reduced precision in anomaly localization [32]. Second, since the current model is primarily trained on public surveillance datasets, transferring it to environments with low illumination, extreme weather, or substantial viewpoint changes may result in false alarms due to feature distribution shifts. Third, the network does not explicitly model the complex interactions between multiple agents, which limits its ability to detect low-frequency or fine-grained anomalous behaviors.

To address these challenges, future work will focus on systematically improving the noise-robust learning mechanisms, enhancing cross-domain generalization, and strengthening the characterization of fine-grained anomalies. We also plan to further refine the model architecture to reduce parameter count and memory consumption, thereby enabling higher frame-rate real-time inference on edge devices. Additionally, we intend to incorporate self-supervised and active learning strategies to decrease reliance on manual annotations and increase robustness against noisy labels. Moreover, we will explore the construction or adoption of synthetic anomaly datasets to systematically evaluate model performance in virtual environments and conduct comparative analyses with real-world results.

## Figures and Tables

**Figure 1 sensors-25-04884-f001:**
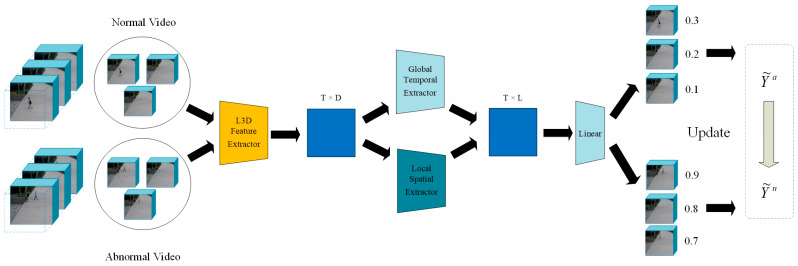
Overview of the model architecture. The training procedure follows these consecutive steps: (1) I3D feature extractor; (2) Local Temporal and Global Spatial Network; (3) dynamic labels and loss function.

**Figure 2 sensors-25-04884-f002:**
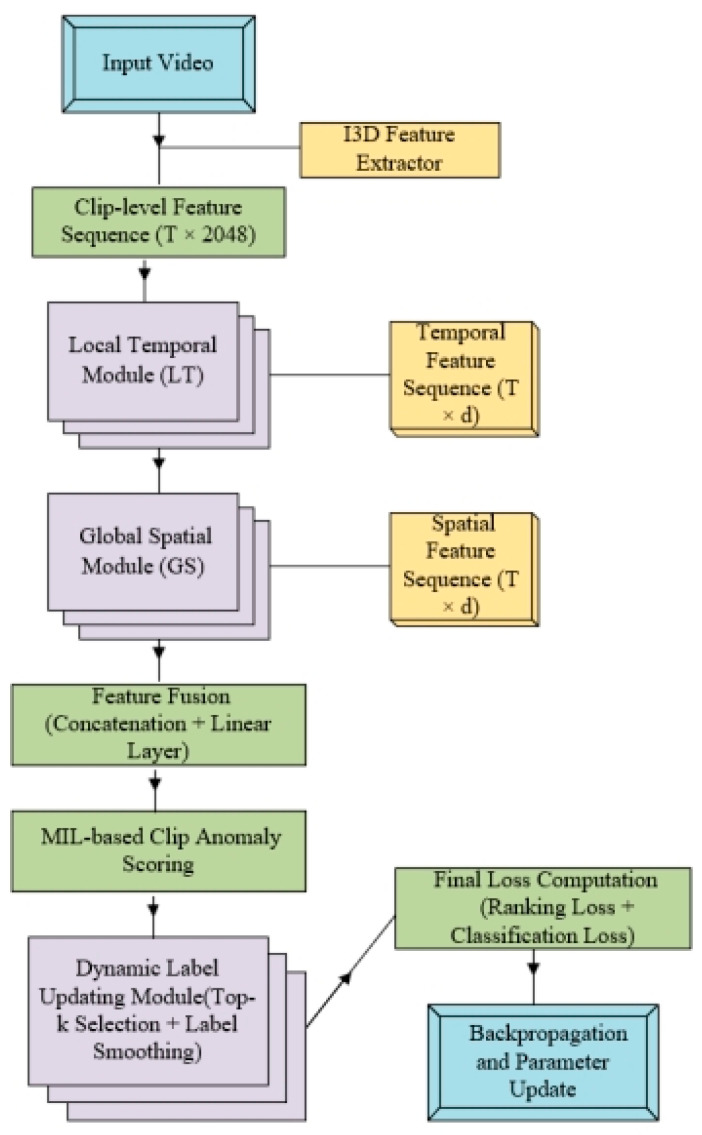
The description of the overall framework of the LTGS network includes three key steps: local temporal relationship module, global spatial module, and dynamic labels.

**Figure 3 sensors-25-04884-f003:**
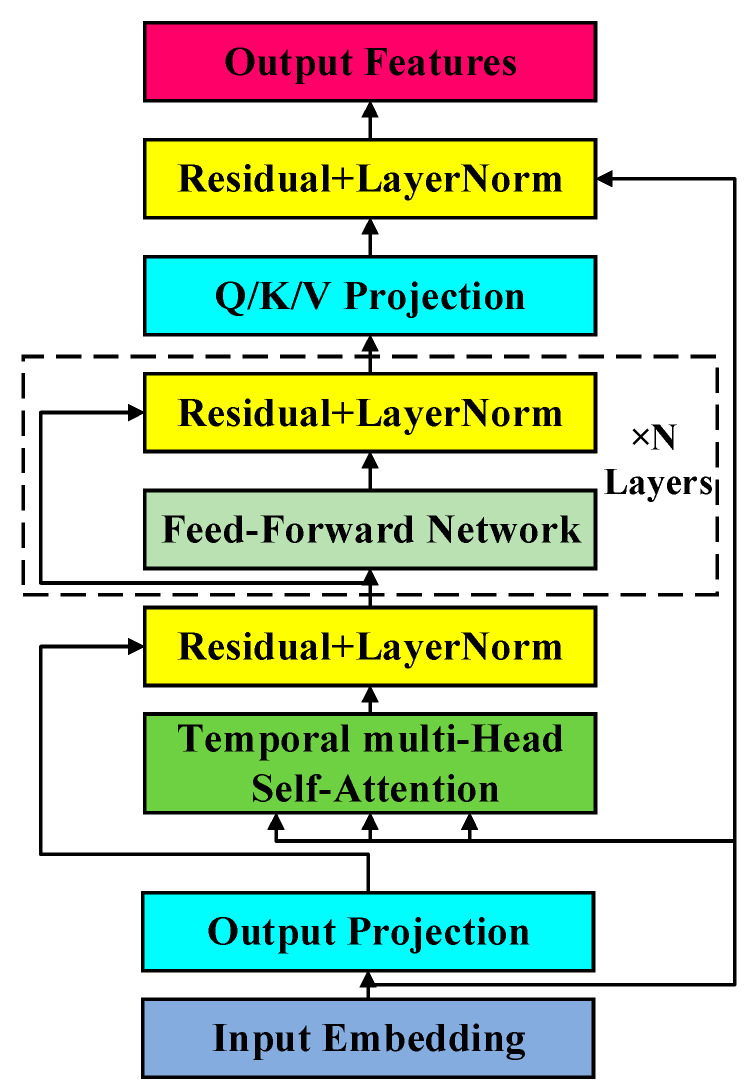
Local temporal module.

**Figure 4 sensors-25-04884-f004:**
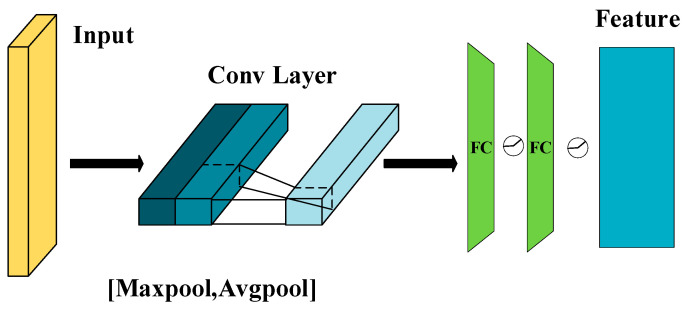
Global spatial module.

**Table 1 sensors-25-04884-t001:** Performance comparisons of Shanghai technology.

Method	Supervised	Encoder	AUC (%)
Conv-AE [27]	Un	-	60.85
VEC [28]	Un	-	74.80
HF2-VAD [29]	Un	-	76.26
GCN-Anomaly [29]	Weak	TSN-RGB	84.44
Zhang et al. [26]	Weak	I3D-RGB	82.50
Sultani et al. [7]	Weak	I3D-RGB	85.33
MIST [16]	Weak	I3D-RGB	92.24
LT	Weak	I3D-RGB	95.96
LTGS	Weak	I3D-RGB	96.55
LTGS + DL	Weak	I3D-RGB	96.69

**Table 2 sensors-25-04884-t002:** Performance comparisons of UCF.

Method	Supervised	Encoder	AUC (%)
SVM	Un	-	50.00
Lu et al. [22]	Un	C3D-RGB	65.51
BODS [30]	Un	I3D-RGB	68.26
GODS [30]	Un	I3D-RGB	70.46
Liu et al. [31]	Full	NLN-RGB	82.00
GCN-Anomaly [12]	Weak	TSN-RGB	82.12
RTFM [17]	Weak	I3D-RGB	82.25
LTGS	Weak	I3D-RGB	82.33

**Table 3 sensors-25-04884-t003:** Ablation study on LTGS-net modules.

Variants	ShanghaiTech AUC (%)	UCF-Crime AUC (%)
Full	96.7	82.3
w/o LT	93.2	79.0
w/o GS	95.1	80.4
w/o Dyn	96.0	80.8
GS→Avg	96.4	81.7

**Table 4 sensors-25-04884-t004:** Ablation study on key mechanisms within LTGS-Net.

Variants	ShanghaiTech AUC (%)	UCF-Crime AUC (%)
Full	96.7	82.3
LT w/o Decay	95.8	81.1
LT(1 Head)	96.2	81.7
w/o Sparse	95.9	81.0
w/o Smooth	95.6	80.9
Dyn(Hard α = 1)	95.7	80.6

**Table 5 sensors-25-04884-t005:** Composition and statistics of the Mini-Anomaly dataset.

Category	Number of Anomaly Videos	Number of Normal Videos	Average Duration (s)
Fighting	10	12	17.5
Illegal Intrusion	9	12	16.8
Abandoned Objects	8	11	18.1
Wrong-way Vehicles	7	14	15.3
Fire and Smoke	6	13	14.9
Total	40	62	16.6

**Table 6 sensors-25-04884-t006:** Performance comparisons of Mini-Anomaly.

Method	Mini-Anomaly AUC (%)
Conv-AE [27]	71.3
GCN-Anomaly [29]	74.8
RTFM [17]	78.5
LTGS-Net	85.6

## Data Availability

The dataset for this experiment came from Wuhan University. All relevant data are within the paper.

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
