# Peer review of "LTGS-Net: Local Temporal and Global Spatial Network for Weakly Supervised Video Anomaly Detection"

_sensors, 2025, doi:10.3390/s25164884_

Round 1
Reviewer 1 Report (Previous Reviewer 1)
Comments and Suggestions for Authors
The additional tests I requested in previous review round notably enhanced the paper quality, since the proposed method is now evaluated over a wider range of instances. Thank you for having finally addressed all of my comments.
Author Response
Thank you for your constructive feedback and for acknowledging the improvements made in the paper. We are glad to hear that the additional tests have enhanced the quality of the work. Your suggestions were instrumental in helping us refine the evaluation and ensure a more comprehensive analysis. We appreciate your time and effort in reviewing the paper.
Reviewer 2 Report (New Reviewer)
Comments and Suggestions for Authors
- In the conclusion, it is mentioned that it shows stronger detection ability compared to other methods in the context of weak supervised learning. What are the methods, please list them out.
- The image in Figure 1 cannot be previewed; Figure 3 is not clear about the meaning the author wants to express, redraw it.
- Are there any literature references for the formulas and mathematical symbols involved in Methodology? If so, please mark them out.
- The first second paragraphs of the introduction should be merged, and the last sentence of the first paragraph should be placed in the last paragraph.
- Section 2.1 introduces the limitations Reconstruction-Based Unsupervised Methods, what are its advantages, it has not been proposed in the text.
- Delete lines 144, 145, 146.
- Align the formula numbers in lines 254, 261, 267, 291,304, 309.
- There are too few references, please add 5 to 8 more.
- Is formula 5 written correctly
- Merge lines 185 to 207 into a single paragraph.
Author Response
Please see the attachment.

This manuscript is a resubmission of an earlier submission. The following is a list of the peer review reports and author responses from that submission.
Round 1
Reviewer 1 Report
Comments and Suggestions for Authors
See attached file.

Reviewer 2 Report
Comments and Suggestions for Authors
1- abstract: better contextualize the problem and the need for LTGS.
2- How does the dynamic label updating strategy impact training stability over time, and is there evidence of convergence or oscillation across epochs?
3- Can the authors clarify whether the spatial module contributes significantly beyond temporal attention alone, especially given the small AUC improvement?
4- Was any ablation study conducted to assess the sensitivity of LTGS performance to hyperparameters like α and β in dynamic label updates?
5- How generalizable is the LTGS architecture to other datasets or real-world scenarios not covered by ShanghaiTech and UCF-Crime?
6- Add a framework with all the steps of the proposed methodology
Reviewer 3 Report
Comments and Suggestions for Authors
This paper proposes a weakly-supervised video anomaly detection method based on the Local Temporal and Global Spatial Network (LTGS-Net), which significantly improves detection accuracy by integrating spatio-temporal features and a dynamic label refinement strategy. The core innovations include: designing a Local Temporal (LT) module that employs a multi-head attention mechanism to capture long- and short-term temporal dependencies between video clips, combined with a sine function to quantify temporal distance weights; proposing a Global Spatial (GS) module that extracts scene-level spatial features through adaptive fusion of average pooling and max pooling; introducing a dynamic label update mechanism to mitigate label noise caused by propagating video-level labels to clip-level annotations via joint optimization of multi-instance ranking loss and classification loss. Experiments demonstrate that the method achieves AUC values of 96.69% and 82.33% on the ShanghaiTech and UCF-Crime datasets, respectively, outperforming state-of-the-art methods (e.g., surpassing MIST by 4.45% on ShanghaiTech). LTGS-Net provides an effective spatio-temporal fusion framework for weakly-supervised anomaly detection, but its methodological transparency, experimental rigor, and practical deployment feasibility require further validation and optimization.
- Figure 1 appears blurred when enlarged and is suspected to be a screenshot. The authors are advised to redraw this figure for improved clarity.
- It is recommended that the authors add a section in the Introduction to outline the content framework and structural organization of the paper.
- The formatting of equations should be unified. Descriptions of parameters in equations (e.g., Equations 1 and 2) should avoid indentation.
- In all experiments, the value of k is set to 3, but the rationale for this parameter choice lacks theoretical analysis or empirical justification.
- The experimental setup (e.g., hardware configuration, software environment) is not described in the Experiments section, which hinders reproducibility.
- In Table 1, methods such as Conv-AE and HF2-VAD lack proper citations. To ensure academic rigor, all compared methods must be explicitly referenced.
- In Table 2, the analysis focuses primarily on the superiority of the proposed LTGS method, while the performance of baseline methods is insufficiently discussed. A detailed comparative analysis is required to validate the effectiveness of the proposed approach.
- The dynamic label update procedure in Algorithm 1 is described too briefly. Key steps lack mathematical explanations, and the selection criteria for input parameters are not provided. Additionally, ablation studies are needed to quantify the contribution of dynamic labels to performance.
- The analogy between video clip features and word vectors in natural language processing is inadequately justified. The inherent differences between video clips and text sequences are not addressed.
- The formatting of mathematical symbols in equations should align with the rest of the manuscript to enhance readability.
Round 2
Reviewer 1 Report
Comments and Suggestions for Authors
The paper notably improved after its revision. However, I deem it still has some points needing to be properly revised.
Minor comments:
- The Details of MIL and RTFM that the Authors explained in the cover letter should be added in the text: readers will surely appreciate them.
- In the same vien, the explanations related to k and β should be added in the text.
Major comment:
- I deem that addiotanal tests must be performed at this stage, rather than during future works. Indeed, by analyzing the relative results more insight can be defined, thus (possibly) enhancing the pace quality. Therefore, I copy-paste the relative comment of the previous review report.
I suggest an additional test in order to assess the accuracy of the system. It would be interesting to understand how the system would classify anomalous scenes, similar to those present in the exploited datasets, which are simulated, rather than actually occurring. I invite the Authors to collect an additional dataset, even a small one, containing such examples, and to report the relative classification results.
Reviewer 3 Report
Comments and Suggestions for Authors
The authors have solved all the comments, it can be accepted.
Author Response
Comment: The authors have solved all the comments, it can be accepted.
Response: Thank you very much for your positive evaluation and for confirming that our revisions have addressed all prior concerns. We appreciate the time and effort you devoted to reviewing our manuscript.